# Parameterised Enumeration for Modification Problems[†]

**Nadia Creignou [1], Raïda Ktari [2], Arne Meier [3,\*] , Julian-Steffen Müller [3], Frédéric Olive [1] and Heribert Vollmer [3]**

[1] Aix-Marseille Université, CNRS, LIS, 13003 Marseille, France
[2] Pôle technologique de Sfax, Université de Sfax, Sfax 3000, Tunisia
[3] Institut für Theoretische Informatik, Leibniz Universität Hannover, 30167 Hannover, Germany
[\*] Correspondence: meier@thi.uni-hannover.de; Tel.: +49-(0)511-762-19768
[†] This paper is an extended version of our paper published in Parameterized Enumeration for Modification Problems. In Proceedings of the Language and Automata Theory and Applications—9th International Conference, Nice, France, 2–6 March 2015.

**Abstract:** Recently, Creignou et al. (Theory Comput. Syst. 2017), introduced the class DelayFPT into parameterised complexity theory in order to capture the notion of efficiently solvable parameterised enumeration problems. In this paper, we propose a framework for parameterised ordered enumeration and will show how to obtain enumeration algorithms running with an FPT delay in the context of general modification problems. We study these problems considering two different orders of solutions, namely, lexicographic order and order by size. Furthermore, we present two generic algorithmic strategies. The first one is based on the well-known principle of self-reducibility and is used in the context of lexicographic order. The second one shows that the existence of a neighbourhood structure among the solutions implies the existence of an algorithm running with FPT delay which outputs all solutions ordered non-decreasingly by their size.

**Keywords:** parameterised complexity; enumeration; bounded search tree; parameterised enumeration; ordering

## 1. Introduction

Given a computational problem one often is interested in generating all solutions. For instance, one wants to list all answers to a query to a database [1] or is interested in all hits that a web search engine produces [2]. Even in bioinformatics [3] or computational linguistics [4] such enumeration problems play a crucial role. In this setting, one is more interested in the delay between output solutions rather than in the overall runtime of such algorithms. Here, a uniform stream of solutions is highly desired. Johnson et al. [5] explain in their seminal paper that the notion of the complexity class DelayP, which consists of problems whose delay is bounded by a polynomial in the input length, is very important.

A view on studied enumeration problems fuels the observation that often a specific order in the output solutions is very central: Many applications benefit from printing "cheap" solutions first. Moreover, enumerating all solutions in non-decreasing order allows to determine not only the smallest solution, but also the $k$th-smallest one. Such a generating algorithm allows finding the smallest solution obeying further constraints (at each generation step one verifies which constraints match). Unfortunately, this technique cannot guarantee efficient enumeration because a long prefix of candidates may not satisfy them. Yet, this technique is very versatile due to its applicability to any additional decidable constraint [6]. Now, we want to exemplify this observation.

Creignou and Hébrard [7] studied, within the well-known Schaefer framework for Boolean constraint satisfaction problems [8], which classes of propositional CNF formulas enumerating all satisfying solutions is possible in DelayP. They showed that for the classes of Horn, anti-Horn, affine or bijunctive formulas, such an algorithm exists. However, for any other class of formulas, having a DelayP algorithm implies P = NP. Interestingly, their proof builds on the self-reducibility of the propositional satisfiability problem. By the approach of a flashlight search, that is, trying first an assignment 0 and then 1, they observed that their enumeration algorithm obeys lexicographic order.

Later, Creignou et al. [9] studied enumerating satisfying assignments for propositional formulas in non-decreasing weight. Surprisingly, now, efficiently enumerating is only possible for Horn formulas and width-2 affine formulas (that is, affine formulas with at most two literals per clause). To achieve their result, the authors exploited priority queues to ensure enumeration in order (as was observed already by Johnson et al. [5]).

While parameterised enumeration had already been considered before (see, e.g., the works of Fernau, Damaschke and Fomin et al. [10–12]), the notion of fixed-parameter tractable delay was novel, leading to the complexity class DelayFPT [13]. Intuitively, the "polynomial time" in the definition of DelayP here is substituted by a fixed-parameter runtime-bound of the form $n^{O(1)} \cdot f(k)$, where $n$ denotes the input length, $k$ is the input parameter and $f$ is a computable function. This introduces the notion of efficiency in the context of the parameterised world, that is, fixed-parameter tractability (FPT), to the enumeration framework. Creignou et al. [13] investigated a wealth of problems from propositional logic and developed enumeration algorithms based on self-reducibility and on the technique of kernelisation. Particularly, the membership of an enumeration problem in DelayFPT can be characterised by a specificly tailored form of kernelisability, very much as in the context of usual decision problems.

As this area of parameterised enumeration is rather young and has received less attention, we want to further support this topic with this paper. Here, we study ordered enumeration in the context of parameterised complexity. First, we introduce arbitrary orders to the parameterised enumeration framework. Then we consider the special context of graph modification problems where we are interested in ordered enumeration for the two mostly studied orders, namely by lexicographic and by non-decreasing size (where the size is the number of modifications that have to be made). We use two algorithmic strategies, depending on the respective order as follows. Based on the principle of self-reducibility we obtain DelayFPT (and polynomial-space) enumeration algorithms for lexicographic order, as soon as the decision problem is efficiently solvable. Secondly, we present a DelayFPT enumeration algorithm for order by size as soon as a certain FPT-computable neighbourhood function on the solutions set exists (see Theorem 1). Notice that the presented algorithms do not enumerate the set of minimal solutions but the set of solutions of bounded size. Extending to such solutions from minimal ones in the enumeration process is not generally trivial. To cope with the order, we use a priority queue that may require exponential space in the input length (as there exist potentially that many solutions).

Eventually, we show that the observed principles and algorithmic strategies can be applied to general modification problems as well. For instance, a general modification problem could allow to flip bits of a string. It is a rather rare situation that a general algorithmic scheme is developed. Usually algorithms are devised on a very individual basis. We prove a wide scope of applicability of our method by presenting new FPT delay ordered enumeration algorithms for a large variety of problems, such as cluster editing [14], triangulation [15], triangle deletion [16], closest-string [17] and backdoor sets [18]. Furthermore, there already exists work which adopts the introduced framework of Creignou et al. [13] in the area of conjunctive query enumeration [19], triangle enumeration [20], combinatorial optimisation [21], abstract argumentation [22] and global constraints [23].

## 2. Preliminaries

We start by defining parameterised enumeration problems with a specific ordering and their corresponding enumeration algorithms. Most definitions in this section transfer those of Johnson et al. and Schmidt [5,24] from the context of enumeration and those of Creignou et al. [13] from the context of parameterised enumeration to the context of parameterised ordered enumeration.

The studied orderings of enumeration problems in this paper are quasi-orders which will be defined in the following.

**Definition 1** (Quasi-Order). *Let R be a set and $\preceq$ a binary relation on R. Then $\preceq$ is a preorder (or quasi-order) if we have for all elements $a, b, c \in R$:*

- *$a \preceq a$ and*
- *if $a \preceq b$ and $b \preceq c$ then $a \preceq c$.*

We will write $z \not\preceq y$ whenever $z \preceq y$ is not true.

Now, we proceed by introducing parameterised enumeration problems with ordering. Intuitively, the corresponding enumeration algorithm for such problems has to obey the given ordering, that is, it has to produce solutions without violating that ordering.

**Definition 2.** *A parameterised enumeration problem with ordering is a quadruple $E = (I, \kappa, \mathrm{Sol}, \preceq)$ such that the following holds:*

- *I is the set of instances.*
- *$\kappa \colon I \to \mathbb{N}$ is the parameterisation function; $\kappa$ is required to be polynomial time computable.*
- *$\mathrm{Sol}$ is a function such that for all $x \in I$, $\mathrm{Sol}(x)$ is a finite set, the set of solutions of x. Further we write $\mathcal{S} = \bigcup_{x \in I} \mathrm{Sol}(x)$.*
- *$\preceq$ is a quasi-order on $\mathcal{S}$.*

Notice that this order on all solutions is only a short way of simultaneously giving an order for each instance. Furthermore, we will write an index *E* letter, e.g., $I_E$, $\kappa_E$, to denote that we are talking about an instance set, parameterisation function, etc., of a given enumeration problem *E*. In the next step, we fix the notion of enumeration algorithms in our setting.

**Definition 3** (Enumeration Algorithm). *Let $E = (I, \kappa, \mathrm{Sol}, \preceq)$ be a parameterised enumeration problem with ordering. Then an algorithm $\mathcal{A}$ is an enumeration algorithm for E if the following holds:*

- *For every $x \in I$, $\mathcal{A}(x)$ terminates after a finite number of steps.*
- *For every $x \in I$, $\mathcal{A}(x)$ outputs exactly the elements of $\mathrm{Sol}(x)$ without duplicates.*
- *For every $x \in I$ and $y, z \in \mathrm{Sol}(x)$, if $y \preceq z$ and $z \not\preceq y$ then $\mathcal{A}(x)$ outputs solution y before solution z.*

Before we define complexity classes for parameterised enumeration, we need the notion of delay for enumeration algorithms.

**Definition 4** (Delay). *Let $E = (I, \kappa, \mathrm{Sol}, \preceq)$ be a parameterised enumeration problem with ordering and $\mathcal{A}$ be an enumeration algorithm for E. Let $x \in I$ be an instance. The ith delay of $\mathcal{A}$ is the elapsed runtime with respect to $|x|$ of $\mathcal{A}$ between outputting the ith and $(i+1)$th solution in $\mathrm{Sol}(x)$. The 0th delay is the precomputation time which is the elapsed runtime with respect to $|x|$ of $\mathcal{A}$ from the start of the computation to the first output statement. Analogously, the nth delay, for $n = |\mathrm{Sol}(x)|$, is the postcomputation time, which is the elapsed runtime with respect to $|x|$ of $\mathcal{A}$ after the last output statement until $\mathcal{A}$ terminates. Then, the delay of $\mathcal{A}$ is the maximum over all $0 \le i \le n$ of the ith delay of $\mathcal{A}$.*

Now we are able to define two different complexity classes for parameterised enumeration following the notion of Creignou et al. [13].

**Definition 5.** *Let $E = (I, \kappa, \text{Sol}, \preceq)$ be a parameterised enumeration problem. We say that E is* FPT *enumerable if there exists an enumeration algorithm $\mathcal{A}$, a computable function $f : \mathbb{N} \to \mathbb{N}$ and a polynomial p such that for every $x \in I$, $\mathcal{A}$ outputs all solutions of $\text{Sol}(x)$ in time $f(\kappa(x)) \cdot p(|x|)$.*

*An enumeration algorithm $\mathcal{A}$ is a DelayFPT algorithm if there exists a computable function $f : \mathbb{N} \to \mathbb{N}$ and a polynomial p such that for every $x \in I$, $\mathcal{A}$ outputs all solutions of $\text{Sol}(x)$ with delay of at most $f(\kappa(x)) \cdot p(|x|)$.*

*The class* DelayFPT *consists of all parameterised enumeration problems that admit a DelayFPT-enumeration algorithm.*

Some of our enumeration algorithms will make use of priority queues to enumerate all solutions in the correct order and to avoid duplicates. We will follow the approach of Johnson et al. [5]. For an instance *x* of a parameterised enumeration problem whose sizes of solutions are polynomially bounded in $|x|$; we use a priority queue $Q$ to store a subset of $\text{Sol}(x)$ of cardinality potentially exponential in $|x|$. The insert operation of $Q$ requires $O(|x| \cdot \log |\text{Sol}(x)|)$ time. The extract minimum operation requires $O(|x| \cdot \log |\text{Sol}(x)|)$ time, too. It is important, however, that the computation of the order between two elements takes at most $O(|x|)$ time. As pointed out by Johnson et al., the required queue can be implemented with the help of standard balanced tree schemes [25].

### 2.1. Graph Modification Problems

Graph modification problems have been studied for a long time in computational complexity theory [26]. Already in the monograph by Garey and Johnson [27], among the graph-theoretic problems considered, many fall into this problem class. To the best of our knowledge, graph modification problems were studied in the context of parameterised complexity for the first time in [28].

In this paper, we consider only undirected graphs. Let $\mathcal{G}$ denote the set of all undirected graphs. A graph property $\mathcal{P} \subseteq \mathcal{G}$ is a set of graphs. Given a graph property $\mathcal{P}$ and an undirected graph $G$, we write $G \models \mathcal{P}$ if the graph $G$ obeys the property $\mathcal{P}$, that is, $G \in \mathcal{P}$.

**Definition 6** (Graph Operations). *A graph operation for G is either of the following:*

- *removing a vertex: A function $\text{rem}_v : \mathcal{G} \to \mathcal{G}$ such that $\text{rem}_v(G)$ is the graph obtained by removing the vertex v from G (if v is present; otherwise $\text{rem}_v$ is the identity) and deleting all incident edges to v,*
- *adding/removing an edge: A function $\text{add}_{\{u,v\}}, \text{rem}_{\{u,v\}} : \mathcal{G} \to \mathcal{G}$ such that $\text{add}_{\{u,v\}}(G), \text{rem}_{\{u,v\}}(G)$ is the graph obtained by adding/removing the edge $\{u, v\}$ to G if u and v are present in G; otherwise both functions are the identity*

*Two operations $o, o'$ are dependent if*

- *$o = \text{rem}_v$ and $o' = \text{rem}_{\{u,v\}}$ (o removes the vertex v and $o'$ removes an edge incident to v) or*
- *$o = \text{rem}_{\{u,v\}}$ and $o' = \text{add}_{\{u,v\}}$ (o removes the edge $\{u, v\}$ and $o'$ adds the same edge $\{u, v\}$ again).*

*A set of operations is consistent if it does not contain two dependent operations. Given such a consistent set of operations S, the graph obtained from G by applying the operations in S on G is denoted by $S(G)$.*

Now, we turn towards the definition of solutions and will define minimality in terms of being inclusion-minimal.

**Definition 7** (Solutions). *Given a graph property $\mathcal{P}$, a graph G, $k \in \mathbb{N}$ and a set of operations O, we say that S is a solution for $(G, k, O)$ with respect to $\mathcal{P}$ if the following three properties hold:*

1. *$S \subseteq O$ is a consistent set of operations,*
2. *$|S| \leq k$ and*
3. *$S(G) \models \mathcal{P}$.*

*A solution S is minimal if there is no solution $S'$ such that $S' \subsetneq S$.*

Cai [28] was interested in the following parameterised graph modification decision problem with respect to a given graph property $\mathcal{P}$:

| Problem: | $\mathcal{M}_{\mathcal{P}}$ |
|---|---|
| Input: | $(G, k, O)$, $G$ undirected graph, $k \in \mathbb{N}$, $O$ set of operations on $G$. |
| Parameter: | The integer $k$. |
| Question: | Does there exist a solution for $(G, k, O)$ with respect to $\mathcal{P}$? |

Some of the most important examples of graph modification problems are presented now. A chord in a graph $G = (V, E)$ is an edge between two vertices of a cycle $C$ in $G$ which is not part of $C$. A given graph $G = (V, E)$ is triangular (or chordal) if each of its induced cycles of four or more nodes has a chord. The problem TRIANGULATION then asks, given an undirected graph $G$ and $k \in \mathbb{N}$, whether there exists a set of at most $k$ edges such that adding this set of edges to $G$ makes it triangular. Yannakakis showed that this problem is NP complete [15]. Kaplan et al. [29] and independently Cai [28] have shown that the parameterised problem is in FPT. For this problem, a solution is a set of edges which have to be added to the graph to make the graph triangular. Observe that, in this special case of the modification problem, the underlying property $\mathcal{P}$, "to be triangular", does not have a finite forbidden set characterisation (since cycles of any length are problematic). Nevertheless, we will see later, that one can efficiently enumerate all minimal solutions as well.

A cluster is a graph such that all its connected components are cliques. In order to transform (or modify) a graph $G$ we allow here only two kinds of operations: Adding or removing an edge. CLUSTER-EDITING asks, given a graph $G$ and a parameter $k$, whether there exists a consistent set of operations of cardinality at most $k$ such that $S(G)$ is cluster. It was shown by Shamir et al., that the problem is NP complete [14].

The problem TRIANGLE-DELETION asks whether a given graph can be transformed into a triangle-free graph by deletion of at most $k$ vertices. Yannakakis has shown that the problem is NP complete [16].

Analogous problems can be defined for many other classes of graphs, e.g., line graphs, claw-free graphs, Helly circular-arc graphs, etc., see [30].

Now, we turn towards the main focus of the paper. Here, we are interested in corresponding enumeration problems with ordering. In particular, we will focus on two well-known preorders, lexicographic ordering and ordering by size. Since our solutions are subsets of an ordered set of operations, they can be encoded as binary strings in which the $i$th bit from right indicates whether the $i$th operation is in the subset. We define the lexicographic ordering of solutions as the lexicographic ordering of these strings. Then, the size of a solution simply is its cardinality.

| Problem: | ENUM-$\mathcal{M}_{\mathcal{P}}^{\text{LEX}}$ |
|---|---|
| Input: | $(G, k, O)$, $G$ undirected graph, $k \in \mathbb{N}$, $O$ ordered set of operations on $G$. |
| Parameter: | The integer $k$. |
| Output: | All solutions of $(G, k, O)$ with respect to $\mathcal{P}$ in lexicographic order. |

| Problem: | ENUM-$\mathcal{M}_{\mathcal{P}}^{\text{SIZE}}$ |
|---|---|
| Input: | $(G, k, O)$, $G$ undirected graph, $k \in \mathbb{N}$, $O$ set of operations on $G$. |
| Parameter: | The integer $k$. |
| Output: | All solutions of $(G, k, O)$ with respect to $\mathcal{P}$ in non-decreasing size. |

If the context is clear, we omit the subscript $\mathcal{P}$ for the graph modification problem and simply write $\mathcal{M}$. Furthermore, we write $\text{Sol}_{\mathcal{M}}(x)$ for the function associating solutions to a given instance, and also $\mathcal{S}_{\mathcal{M}}$ for the set of all solutions of $\mathcal{M}$.

## 3. Enumeration of Graph Modification Problems with Ordering

In this section, we study the two previously introduced parameterised enumeration problems with ordering (lexicographic and size ordering).

### 3.1. Lexicographic Ordering

We first prove that, for any graph property $\mathcal{P}$, if the decision problem $\mathcal{M}_{\mathcal{P}}$ is in FPT then there is an efficient enumeration algorithm for ENUM-$\mathcal{M}_{\mathcal{P}}^{\text{LEX}}$.

**Lemma 1.** *Let $\mathcal{M}_{\mathcal{P}}$ be a graph modification problem. If $\mathcal{M}_{\mathcal{P}}$ is in* FPT *then* ENUM-$\mathcal{M}_{\mathcal{P}}^{\text{LEX}} \in$ DelayFPT *with polynomial space.*

**Proof.** Algorithm 1 enumerates all solutions of an instance of a given modification problem $\mathcal{M}_{\mathcal{P}}$ by the method of self-reducibility (it is an extension of the flash light search of Creignou and Hébrard [7]). The algorithm uses a function ExistsSol$(G, k, O)$ that tests if the instance $(G, k, O)$ of the modification problem $\mathcal{M}_{\mathcal{P}}$ has a solution. By the assumption of the lemma, $\mathcal{M}_{\mathcal{P}} \in$ FPT so this function runs in FPT time. We use calls to this function to avoid exploration of branches of the recursion tree that do not lead to any output. Moreover, we ensure that the solutions using $o_p$ have to be consistent. This consistency check runs in polynomial time for graph operations. The rest yields a search tree of depth at most $k$. From this it follows that, for any instance of length $n$, the time beween the output of any two solutions is bounded by $f(k) \cdot p(n)$ for some polynomial $p$ and a computable function $f$. □

---
**Algorithm 1:** Enumerate all solutions of $\mathcal{M}_{\mathcal{P}}$ in lexicographic order

---
    **Input:** $(G, k, O)$: A graph $G$, $k \in \mathbb{N}$, an ordered set of operations $O = \{o_1, \ldots, o_n\}$
    **Output:** all consistent sets $S \subseteq O$ s.t. $|S| \leq k$, $S(G) \models \mathcal{P}$ in lexicographic order
1 **if** ExistsSol$(G, k, O)$ **then** Generate$(G, k, O, \emptyset)$;

    **Procedure** Generate$(G, k, O, S)$:
1 **if** $O = \emptyset$ *or* $k = 0$ **then return** $S$;
2 **else**
3      let $o_p$ be the lexicographically last operation in $O$, $O' := O \setminus \{o_p\}$;
4      **if** ExistsSol$(S(G), k, O')$ **then** Generate$(S(G), k, O', S)$;
5      **if** $S \cup \{o_p\}$ *is consistent and* ExistsSol$((S \cup \{o_p\})(G), k - 1, O')$ **then**
6          Generate$((S \cup \{o_p\})(G), k - 1, O', S \cup \{o_p\})$.

---

**Corollary 1.** ENUM-TRIANGULATION$^{\text{LEX}} \in$ DelayFPT *with polynomial space.*

**Proof.** Kaplan et al. [29] and Cai [28] showed that TRIANGULATION $\in$ FPT. Now, by applying Lemma 1, we get the result. □

Cai [28] identified a class of graph properties whose associated modification problems belong to FPT. Let us introduce some terminology.

**Definition 8.** *Given two graphs $G = (V, E)$ and $H = (V', E')$, we write $H \trianglelefteq G$ if $H$ is an induced subgraph of $G$, i.e., $V' \subseteq V$ and $E' = E \cap (V' \times V')$. Let $\mathcal{F}$ be a set of graphs and $\mathcal{P}$ be a graph property. We say that $\mathcal{F}$ is a forbidden set characterisation of $\mathcal{P}$ if for any graph $G$ it holds that: $G \models \mathcal{P}$ iff for all $H \in \mathcal{F}, H \ntrianglelefteq G$.*

Among the problems presented in the previous section (see page 5), TRIANGLE-DELETION and CLUSTER-EDITING have a finite forbidden set characterisation, namely by triangles and paths of length two. In contrast to that, TRIANGULATION has a forbidden set characterisation which is infinite, since cycles of arbitrary length are problematic. Actually, for properties having a finite forbidden set

characterisation, the corresponding modification problem is fixed-parameter tractable. Together with Lemma 1, this provides a positive result in terms of enumeration.

**Proposition 1** ([28])**.** *If a property $\mathcal{P}$ has a finite forbidden set characterisation then $\mathcal{M}_\mathcal{P}$ is in* FPT.

**Corollary 2.** *For any graph modification problem, if $\mathcal{P}$ has a finite forbidden set characterisation then* ENUM-$\mathcal{M}_\mathcal{P}^{\text{LEX}} \in$ DelayFPT *with polynomial space.*

**Proof.** This result follows by combining Proposition 1 with Lemma 1.    □

*3.2. Size Ordering*

A common strategy in the enumeration context consists of defining a notion of a neighbourhood that allows to compute a new solution from a previous one with small amounts of computation time (see, e.g., the work of Avis and Fukuda [31]). We introduce the notion of a neighbourhood function, which, roughly speaking, generates some initial solutions from which all solutions can be produced. A priority queue then takes care of the ordering and avoids duplicates, which may require exponential space. For the graph modification problems of interest, we show that if the inclusion-minimal solutions can be generated in FPT, then such a neighbourhood function exists, accordingly providing a DelayFPT-enumeration algorithm. In the following, $\mathbb{O}$ (the "seed") is a technical symbol that will be used to generate the initial solutions.

**Definition 9.** *Let $\mathcal{M}$ be a graph modification problem. A neighbourhood function for $\mathcal{M}$ is a (partial) function $\mathcal{N}_\mathcal{M} \colon I_\mathcal{M} \times (\mathcal{S}_\mathcal{M} \cup \{\mathbb{O}\}) \to 2^{\mathcal{S}_\mathcal{M}}$ such that the following holds:*

1. *For all $x = (G, k, O) \in I_\mathcal{M}$ and $S \in \text{Sol}_\mathcal{M}(x) \cup \{\mathbb{O}\}$, $\mathcal{N}_\mathcal{M}(x, S)$ is defined.*
2. *For all $x \in I_\mathcal{M}$, $\mathcal{N}_\mathcal{M}(x, \mathbb{O}) = \varnothing$ if $\text{Sol}_\mathcal{M}(x) = \varnothing$, and $\mathcal{N}_\mathcal{M}(x, \mathbb{O})$ is an arbitrary set of solutions otherwise.*
3. *For all $x \in I_\mathcal{M}$ and $S \in \text{Sol}_\mathcal{M}(x)$, if $S' \in \mathcal{N}_\mathcal{M}(x, S)$ then $|S| < |S'|$.*
4. *For all $x \in I_\mathcal{M}$ and all $S \in \text{Sol}_\mathcal{M}(x)$, there exists $p > 0$ and $S_1, \ldots, S_p \in \text{Sol}_\mathcal{M}(x)$ such that (i) $S_1 \in \mathcal{N}_\mathcal{M}(x, \mathbb{O})$, (ii) $S_{i+1} \in \mathcal{N}_\mathcal{M}(x, S_i)$ for $1 \leq i < p$ and (iii) $S_p = S$.*

*Furthermore, we say that $\mathcal{N}_\mathcal{M}$ is* FPT *computable, when $\mathcal{N}_\mathcal{M}(x, S)$ is computable in time $f(\kappa(x)) \cdot poly(|x|)$ for any $x \in I_\mathcal{M}$ and $S \in \text{Sol}_\mathcal{M}(x)$.*

As a result, a neighbourhood function for a problem $\mathcal{M}$ is a function that in a first phase computes from scratch an initial set of solutions (see Definition 9(2)). In many of our applications below, $\mathcal{N}_\mathcal{M}(x, \mathbb{O})$ will be the set of all minimal solutions for $x$. In a second phase these solutions are iteratively extended (see condition (3)), where condition (4) guarantees that we do not miss any solution, as we will see in the next theorem.

**Theorem 1.** *Let $\mathcal{M}$ be a graph modification problem. If $\mathcal{M}$ admits a neighbourhood function $\mathcal{N}_\mathcal{M}$ that is* FPT*-computable, then* ENUM-$\mathcal{M}^{\text{SIZE}} \in$ DelayFPT.

**Proof.** Algorithm 2 outputs all solutions in DelayFPT time. By the definition of the priority queue (recall in particular that insertion of an element is done only if the element is not yet present in the queue) and by the fact that all elements of $\mathcal{N}_\mathcal{M}((G, k, O), S)$ are of bigger size than $S$ by Definition 9(3), it is easily seen that the solutions are output in the right order and that no solution is output twice.

Besides, no solution is omitted. Indeed, given $S \in \text{Sol}_\mathcal{M}(G, k, O)$ and $S_1, \ldots, S_p$ associated with $S$ by Definition 9(4), we prove by induction that each $S_i$ is inserted in $Q$ during the run of the algorithm:

$i = 1$**:** This proceeds from line 2 of the algorithm.
$i > 1$**:** The solution $S_{i-1}$ is inserted in $Q$ by induction hypothesis and hence all elements of $\mathcal{N}_\mathcal{M}((G, k, O), S_{i-1})$, including $S_i$, are inserted in $Q$ (line 5 of Algorithm 2). Consequently, each $S_i$ is inserted in $Q$ and then output during the run. In particular, this is true for $S = S_p$.

---

**Algorithm 2:** DelayFPT algorithm for ENUM-$\mathcal{M}$

   **Input :** $(G, k, O)$ : $G$ is an undirected graph, $k \in \mathbb{N}$, and $O$ is a set of operations.

**1** compute $\mathcal{N}_{\mathcal{M}}((G, k, O), \mathbb{O})$;

**2** insert all elements of $\mathcal{N}_{\mathcal{M}}((G, k, O), \mathbb{O})$ into priority queue $Q$ (ordered by size);

**3** **while** $Q$ *is not empty* **do**

**4**    **extract** the minimum solution $S$ of $Q$ and output it;

**5**    **insert** all elements of $\mathcal{N}_{\mathcal{M}}((G, k, O), S)$ into $Q$;

---

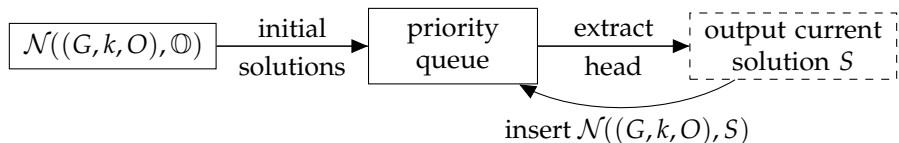

**Figure 1.** Structure of Algorithm 2.

Finally, we claim that Algorithm 2 (see Fig. 1 for a graphical representation) runs in DelayFPT time. Indeed, the delay between the output of two consecutive solutions is bounded by the time required to compute a neighbourhood of the form $\mathcal{N}_{\mathcal{M}}((G, k, O), \mathbb{O})$ or $\mathcal{N}_{\mathcal{M}}((G, k, O), S)$ and to insert all its elements in the priority queue. This is in FPT due to the assumption on $\mathcal{N}_{\mathcal{M}}$ being FPT computable and as there is only a single extraction and many FPT insertion operations in the queue. $\square$

A natural way to provide a neighbourhood function for a graph modification problem $\mathcal{M}$ is to consider the inclusion minimal solutions of $\mathcal{M}$. Let us denote by MIN-$\mathcal{M}$ the problem of enumerating all inclusion minimal solutions of $\mathcal{M}$.

**Theorem 2.** *Let $\mathcal{M}$ be a graph modification problem. If MIN-$\mathcal{M}$ is FPT enumerable then ENUM-$\mathcal{M}^{\text{SIZE}} \in$ DelayFPT.*

**Proof.** Let $\mathcal{A}$ be an FPT algorithm for MIN-$\mathcal{M}$. Because of Theorem 1, it is sufficient to build an FPT-neighbourhood function for $\mathcal{M}$. For an instance $(G, k, O)$ of $\mathcal{M}$ and for $S \in \text{Sol}_{\mathcal{M}}(G, k, O) \cup \{\mathbb{O}\}$, we define $\mathcal{N}_{\mathcal{M}}((G, k, O), S)$ as the result of Algorithm 3.

Accordingly, the function $\mathcal{N}_{\mathcal{M}}$ clearly fulfils Conditions 2 and 3 of Definition 9. We prove by induction that it also satisfies Condition 4 (that is, each solution $T$ of size $k$ comes with a sequence $T_1, \ldots, T_p = T$ such that $T_1 \in \mathcal{N}_{\mathcal{M}}((G, k, O), \mathbb{O})$ and $T_{i+1} \in \mathcal{N}_{\mathcal{M}}((G, k, O), T_i)$ for each $i$). If $T$ is a minimal solution for $(G, k, O)$, then $T \in \mathcal{N}_{\mathcal{M}}((G, k, O), \mathbb{O})$ and the expected sequence $(T_i)_{1 \leq i \leq p}$ reduces to $T_1 = T$. Otherwise, there exists an $S \in \text{Sol}_{\mathcal{M}}(G, k, O)$ and a non-empty set of transformations, say $S' \cup \{t\}$, such that $T = S \cup S' \cup \{t\}$ and there is no solution for $G$ between $S$ and $S \cup S' \cup \{t\}$. This entails that $S'$ is a minimal solution for $((S \cup \{t\})(G), k - |S| - 1)$ and, as a consequence, $T \in \mathcal{N}_{\mathcal{M}}((G, k, O), S)$ (see lines 4–5 of Algorithm 3). The conclusion follows from the induction hypothesis that guarantees the existence of solutions $S_1, \ldots, S_q$ such that $S_1 \in \mathcal{N}_{\mathcal{M}}((G, k, O), \mathbb{O})$, $S_{i+1} \in \mathcal{N}_{\mathcal{M}}((G, k, O), S_i)$ and $S_q = S$. The expected sequence $T_1, \ldots, T_p$ for $T$ is nothing but $S_1, \ldots, S_q, T$. To conclude, it remains to show that Algorithm 3 is FPT. This follows from the fact that $\mathcal{A}$ is an FPT algorithm (Lines 1 and 4 of Algorithm 3). $\square$

**Corollary 3.** ENUM-TRIANGULATION$^{\text{SIZE}} \in$ DelayFPT.

**Proof.** All inclusion-minimal $k$-triangulations can be output in time $O(2^{4k} \cdot |E|)$ for a given graph $G$ and $k \in \mathbb{N}$ as shown by Kaplan et al. [29, Theorem 2.4]. This immediately yields the expected result via Theorem 2. $\square$

---

**Algorithm 3:** Procedure for computing $\mathcal{N}_{\mathcal{M}}((G, k, O), S)$

---

**Input :** $(G, k, O)$, $S$: $G$ is an undirected graph, $k \in \mathbb{N}$, $O$ and $S$ are sets of operations.

1　**if** $S = \mathbb{O}$ **then return** $\mathcal{A}(G, k, O)$;
2　res $:= \varnothing$;
3　**for all** $t \in O$ **do**
4　　**for all** $S' \in \mathcal{A}((S \cup \{t\})(G), k - |S| - 1, O \setminus \{t\})$ **do**
5　　　**if** $S \cup S' \cup \{t\}$ *is consistent* **then** res $:= $ res $\cup \{S \cup S' \cup \{t\}\}$ ;

6　**return** res;

---

**Corollary 4.** *For any property $\mathcal{P}$ that has a finite forbidden set characterisation, the problem* ENUM-$\mathcal{M}_{\mathcal{P}}^{\text{SIZE}}$ *is in* DelayFPT*.*

**Proof.** The algorithm developed by Cai [28] for the decision problem is based on a bounded search tree, whose exhaustive examination provides all size minimal solutions in FPT. Theorem 2 yields the conclusion. □

**Corollary 5.** ENUM-CLUSTER-EDITING$^{\text{SIZE}}$ *and* ENUM-TRIANGLE-DELETION$^{\text{SIZE}}$ *are in* DelayFPT*.*

**Proof.** Both problems have a finite forbidden set characterisation. For the cluster editing problem, paths of length two are the forbidden pattern, and, regarding ENUM-TRIANGLE-DELETION$^{\text{SIZE}}$, the forbidden patterns are triangles. The result then follows from Corollary 4. □

## 4. Generalisation to Modification Problems

In this section, we will show how the algorithmic strategy that has been defined and formalised in the context of graph modification can be of use for many other problems, coming from various combinatorial frameworks.

**Definition 10** (General Operations). *Let $Q \subseteq \Sigma^*$ be a language defined over an alphabet and let $x \in \Sigma^*$ be an input. A set of operations $\Omega(Q) = \{\, \omega_n \colon \Sigma^* \to \Sigma^* \mid n \in \mathbb{N} \,\}$ is an infinite set of operations on instances of $Q$. We say that an operation $\omega$ is valid with respect to an instance $x \in Q$, if $\omega(x) \in Q$. We write $\Omega/x$ for the set of possible (valid) operations on an instance $x$.*

*Two operations $\omega, \omega'$ are dependent with respect to an instance $x \in Q$ if*

- $\omega(\omega'(x)) = x$ or　　　　　　　　　　　　　　　　　*(intuitively, $\omega$ and $\omega'$ cancel out)*
- $\omega(\omega'(x)) = \omega'(x)$ or $\omega(\omega'(x)) = \omega(x)$.　　　*(intuitively, the order of $\omega$ and $\omega'$ does not matter)*

*A set of operations $O \subseteq \Omega/x$ is consistent with respect to $x$ if it does not contain two dependent operations. Similarly, we say that an operation $\omega$ is consistent with a set $S$ if and only if $S \cup \{\omega\}$ is consistent.*

For instance, the set $\Omega$ could contain operations that add edges or, in another case, flip bits. How exactly $\Omega$ is defined highly depends on the corresponding language $Q$.

**Example 1.** *Let $\mathcal{G} \subseteq \{0, 1\}^*$ be the language of all undirected graphs encoded by adjacency matrices. Then $\Omega(\mathcal{G})$ is the set of all graph operations in the sense of Definition 6: Removing vertices or edges, adding edges. Note that $\Omega(\mathcal{G})$ contains all operations of the kind*

$$\text{rem}_i \colon \mathcal{G} \to \mathcal{G}, \quad \text{rem}_{\{i,j\}} \colon \mathcal{G} \to \mathcal{G}, \quad \text{add}_{\{i,j\}} \colon \mathcal{G} \to \mathcal{G}$$

*for all $i, j \in \mathbb{N}$. Furthermore, let $G = (V, E) \in \{0, 1\}^*$ be a concrete input graph. As a result, $\Omega/G$ then is the restriction of $\Omega$ to those $i, j \in \mathbb{N}$ such that $v_i, v_j \in V$ encode vertices in $G$.*

Similarly to how it was defined in Subsection 2.1, a property is just a set. In the following context, it is a subset of a considered language $Q$. Intuitively, in the concept of graph modification problems, one may think of $Q$ as $\mathcal{G}$. Then a graph property $\mathcal{P}$ is just a subset of $\mathcal{G}$.

**Definition 11** (General Solutions). *Let $Q \subseteq \Sigma^*$ be a language defined over an alphabet, $S \subseteq \Omega/x$ be a finite set of operations on $x \in Q$ and $\mathcal{P} \subseteq Q$ be a property. We say that $S$ is a solution (of $x$) if $S$ is a consistent set of operations and $S(x) \in \mathcal{P}$. Furthermore, we denote by $\mathcal{S}_Q := \bigcup_{x \in Q} \{ S \mid S \text{ is a solution of } x \}$ the set of all solutions for every instance $x \in Q$. In addition, $\mathrm{Sol}(x)$ is the set of solutions for every instance $x \in Q$.*

**Example 2.** *Continuing the previous example, if the property $\mathcal{P}$ is "to be a cluster" then a consistent solution $S$ to a given graph is a sequence of removing vertices, adding and deleting of edges where*

- *there is no edge $(i, j)$ added or deleted such that vertex $i$ or $j$ is removed,*
- *there is no edge $(i, j)$ added and removed and*
- *$S(G) \models \mathcal{P}$.*

*Similarly, adding edge $(i, j)$ together with removing vertex $i$ or $j$ or removing edge $(i, j)$ is an inconsistent set of operations.*

Now we want to define the corresponding decision and enumeration tasks. On that account, let $\mathcal{P}$ be a property, $\Pi = (Q, \kappa)$ be a parametrised problem with $Q \subseteq \Sigma^*$ and $\Omega$ be a set of operations.

| Problem: | $\Pi_{\mathcal{P}}$ — parameterised modification problem $\Pi$ w.r.t. a property $\mathcal{P}$ over $\Sigma$ |
|---|---|
| **Input:** | $x \in \Sigma^*$, $k \in \mathbb{N}$, $\Omega/x$ set of operations. |
| **Parameter:** | The integer $k$. |
| **Question:** | Is there a solution $S \subseteq \Omega/x$ and $|S| \leq k$? |

| Problem: | ENUM-MIN-$\Pi_{\mathcal{P}}$ — parameterised minimum enumeration modification problem w.r.t. a property $\mathcal{P}$ over $\Sigma$ |
|---|---|
| **Input:** | $x \in \Sigma^*$, $k \in \mathbb{N}$, $\Omega/x$ set of operations. |
| **Parameter:** | The integer $k$. |
| **Output:** | All minimal (w.r.t. some order) solutions $S \subseteq \Omega/x$ with $|S| \leq k$. |

The enumeration modification problem where we want to output all possible sets of transformations on a given instance $x$ (and not only the minimum ones) then is ENUM-$\Pi_{\mathcal{P}}$.

In the following, we show how the notion of neighbourhood functions can be generalised as well. This will in turn yield generalisations of the results for graph modification problems afterwards.

**Definition 12.** *Let $\Sigma$ be an alphabet, $\mathcal{P} \subseteq \Sigma^*$ be a property and $\Pi_{\mathcal{P}}$ be a parameterised modification problem over $\Sigma$. A neighbourhood function for $\Pi_{\mathcal{P}}$ is a (partial) function $\mathcal{N}_{\Pi_{\mathcal{P}}} \colon \Sigma^* \times (\mathcal{S}_{\Pi_{\mathcal{P}}} \cup \{\mathbb{O}\}) \to 2^{\mathcal{S}_{\Pi_{\mathcal{P}}}}$ such that the following holds:*

1. *For all $x \in \Sigma^*$ and $S \in \mathrm{Sol}_{\Pi_{\mathcal{P}}}(x) \cup \{\mathbb{O}\}$, $\mathcal{N}_{\Pi_{\mathcal{P}}}(x, S)$ is defined.*
2. *For all $x \in \Sigma^*$, $\mathcal{N}_{\Pi_{\mathcal{P}}}(x, \mathbb{O}) = \varnothing$ if $\mathrm{Sol}_{\Pi_{\mathcal{P}}}(x) = \varnothing$, and $\mathcal{N}_{\Pi_{\mathcal{P}}}(x, \mathbb{O})$ is an arbitrary set of solutions otherwise.*
3. *For all $x \in \Sigma^*$ and $S \in \mathrm{Sol}_{\Pi_{\mathcal{P}}}(x)$, if $S' \in \mathcal{N}_{\Pi_{\mathcal{P}}}(x, S)$ then $|S| < |S'|$.*
4. *For all $x \in \Sigma^*$ and all $S \in \mathrm{Sol}_{\Pi_{\mathcal{P}}}(x)$, there exists $p > 0$ and $S_1, \ldots, S_p \in \mathrm{Sol}_{\Pi_{\mathcal{P}}}(x)$ such that (i) $S_1 \in \mathcal{N}_{\Pi_{\mathcal{P}}}(x, \mathbb{O})$, (ii) $S_{i+1} \in \mathcal{N}_{\Pi_{\mathcal{P}}}(x, S_i)$ for $1 \leq i < p$ and (iii) $S_p = S$.*

*Furthermore, we say that $\mathcal{N}_{\Pi_{\mathcal{P}}}$ is FPT computable when $\mathcal{N}_{\Pi_{\mathcal{P}}}(x, S)$ is computable in time $f(k) \cdot poly(|x|)$ for any $x \in \Sigma^*$ and $S \in \mathrm{Sol}_{\Pi_{\mathcal{P}}}(x)$.*

As already announced before, we are able to state generalised versions of Theorems 1 and 2 which can be proven in a similar way. However, one has to replace the graph modification problems by general modification problems.

**Corollary 6.** *Let $\mathcal{P}$ be a property, $\Pi \subseteq \Sigma^* \times \mathbb{N}$ be a parameterised modification problem and $\Omega$ be a set of operations such that $\Omega/x$ is finite for all $x \in \Sigma^*$. If $\Pi_{\mathcal{P}}$ admits a neighbourhood function that is FPT computable then ENUM-$\Pi_{\mathcal{P}} \in$ DelayFPT using*

- *polynomial space for lexicographic order and*
- *exponential space for size order.*

**Corollary 7.** *Let $\mathcal{P}$ be a property, $\Pi \subseteq \Sigma^* \times \mathbb{N}$ be a parameterised modification problem and $\Omega$ be a set of operations such that $\Omega/x$ is finite for all $x \in \Sigma^*$. If ENUM-MIN-$\Pi_{\mathcal{P}}$ is FPT enumerable and the consistency of solutions can be checked in FPT then ENUM-$\Pi_{\mathcal{P}} \in$ DelayFPT and using*

- *polynomial space for lexicographic order and*
- *exponential space for size order.*

*4.1. Closest String*

In the following, we consider a central NP complete problem in coding theory [32]. Given a set of binary strings $I$, we want to find a string $s$ with maximum Hamming distance $\max\{\, d_H(s, s') \mid s' \in I \,\} \leq d$ for a $d \in \mathbb{N}$, where $d_H(s, s')$ is the Hamming distance between two strings.

**Definition 13** (Bit-Flip operation). *Given a string $w = w_1 \cdots w_n$ with $w_i \in \{0, 1\}, n \in \mathbb{N}$ and a set $S \subseteq \{1, \ldots, n\}$, $S(w)$ denotes the string obtained from $w$ by flipping the bits indicated by $S$, more formally $S(w) := S(w_1) \cdots S(w_n)$, where $S(w_i) = 1 - w_i$ if $i \in S$ and $S(w_i) = w_i$ otherwise.*

The corresponding parametrised version is the following.

| Problem: | CLOSEST-STRING |
|---|---|
| Input: | A sequence $(s_1, s_2, ..., s_k)$ of $k$ strings over $\{0, 1\}$ each of given length $n \in N$ and an integer $d \in N$. |
| Parameter: | The integer $d$. |
| Question: | Does there exist $S \subseteq \{1, \ldots, n\}$ such that $d_H(S(s_1), s_i) \leq d$ for all $1 \leq i \leq k$? |

**Proposition 2** ([17]). CLOSEST-STRING *is in* FPT.

Moreover, an exhaustive examination of a bounded search tree constructed from the idea of Gramm et al. [17, Figure 1], allows to produce all minimal solutions of this problem in FPT. Accordingly, we get the following result for the corresponding enumeration problems.

**Theorem 3.**
- ENUM-CLOSEST-STRING$^{\text{LEX}} \in$ DelayFPT *with polynomial space.*
- ENUM-CLOSEST-STRING$^{\text{SIZE}} \in$ DelayFPT *with exponential space.*

**Proof.** $\Omega$ is just the set of operations which flip the $i$th bit of a string for every $i \in \mathbb{N}$. By combining Proposition 2 with Corollary 7 we get the desired result.  □

*4.2. Backdoors*

In this section, we will consider the concept of backdoors. Let $\mathcal{C}$ be a class of propositional formulas. Intuitively, a $\mathcal{C}$ backdoor is a set of variables of a given propositional formula with the following property. Applying assignments over these variables to the formula always yields a formula in the class $\mathcal{C}$. Of course, one aims for formula classes for which satisfiability can be decided efficiently. Informally speaking, with the parameter backdoor size of a formula one tries to describe a distance to tractability. This definition was first introduced by Golmes, Williams and Selman [18] to model short distances to efficient subclasses. Until today, backdoors gained copious attention in many different

areas: abduction [33], answer set programming [34,35], argumentation [36], default logic [37], temporal logic [38], planning [39] and constraint satisfaction [40,41].

Consider a formula $\phi$ in conjunctive normal form. Denote by $\phi[\tau]$ for a partial truth assignment $\tau$ the result of removing all clauses from $\phi$ which contain a literal $\ell$ with $\tau(\ell) = 1$ and removing literals $\ell$ with $\tau(\ell) = 0$ from the remaining clauses.

**Definition 14.** *Let $\mathcal{C}$ be a class of CNF formulas and $\phi$ be a CNF formula. A set $V \subseteq Vars(\phi)$ of variables of $\phi$ is a strong $\mathcal{C}$ backdoor set of $\phi$ if for all truth assignments $\tau \colon V \to \{0,1\}$ we have that $\phi[\tau] \in \mathcal{C}$.*

**Definition 15** ([42,43]). *Let $\mathcal{C}$ be a class of CNF formulas and $\phi$ be a CNF formula. A set $V \subseteq Vars(\phi)$ of variables of $\phi$ is a $\mathcal{C}$-deletion backdoor set of $\phi$ if $\phi[V]$ is in $\mathcal{C}$, where $\phi[V]$ denotes the formula obtained from $\phi$ by deleting in $\phi$ all occurrences of variables from $V$.*

**Definition 16** (Weak Backdoor Sets). *Let $\mathcal{C}$ be a class of CNF formulas and $\phi$ be a propositional CNF formula. A set $V \subseteq Vars(\phi)$ of variables from $\phi$ is a weak $\mathcal{C}$ backdoor set of $\phi$ if there exists an assignment $\theta \in \Theta(V)$ such that $\phi[\theta] \in \mathcal{C}$ and $\phi[\theta]$ is satisfiable.*

Now let us consider the following enumeration problem.

| Problem: | ENUM-WEAK-BACKDOORSET($\mathcal{C}$) |
| --- | --- |
| Input: | A formula $\phi$ in CNF, $k \in \mathbb{N}$. |
| Parameter: | The integer $k$. |
| Output: | The set of all weak $\mathcal{C}$ backdoor sets of $\phi$ of size at most $k$. |

Similarly, define ENUM-STRONG-BACKDOORSET($\mathcal{C}$) for the set of all strong $\mathcal{C}$ backdoor sets of $\phi$ of size at most $k$. Observe that the backdoor set problems can be seen as modification problems where solutions are sequences of variable assignments. The target property then simply is the class of CNF formulas $\mathcal{C}$.

Notice that Creignou et al. [13, Theorem 4], have studied enumeration for exact strong HORNbackdoor sets and provided an algorithm running in DelayFPT, where HORN denotes the set of all Horn formulas, that is, CNF formulas whose clauses contain at most one positive literal.

**Definition 17** (Base Class [44]). *The class $\mathcal{C}$ is a base class if it can be recognised in P (that is, $\mathcal{C} \in$ P), the satisfiability of its formulas is in P and the class is closed under isomorphisms w.r.t. variable names. We say that $\mathcal{C}$ is clause defined if for every CNF formula $\phi$ we have: $\phi \in \mathcal{C}$ if and only if $\{C\} \in \mathcal{C}$ for all clauses C from $\phi$.*

**Proposition 3** ([44, Proposition 2]). *For every clause-defined base class $\mathcal{C}$, the detection of weak $\mathcal{C}$ backdoor sets is in FPT for input formulas in 3-CNF.*

In their proof, Gaspers and Szeider [44] describe how utilising a bounded search tree allows one to solve the detection of weak $\mathcal{C}$ backdoors in FPT time. Interesting to note, this technique results in obtaining all minimal solutions in FPT time. This observation results in the following theorem.

**Theorem 4.** *For every clause-defined base class $\mathcal{C}$ and input formula in 3-CNF*

- ENUM-WEAK-$\mathcal{C}$-BACKDOORS$^{\text{LEX}} \in$ DelayFPT *with polynomial space and*
- ENUM-WEAK-$\mathcal{C}$-BACKDOORS$^{\text{SIZE}} \in$ DelayFPT *with exponential space.*

**Proof.** The set of operations $\Omega$ contains functions that replace a specific variable $i \in \mathbb{N}$ by a truth value $t \in \{0,1\}$. A solution encodes the chosen backdoor set together with the required assignment. Proposition 3 yields ENUM-MIN-WEAK-$\mathcal{C}$BACKDOORS$^{\text{LEX}}$, resp., ENUM-MIN-WEAK-$\mathcal{C}$-BACKDOORS$^{\text{SIZE}}$ being FPT enumerable. As the consistency check for solutions is in polynomial time, applying Corollary 7 completes the proof. □

In the following result, we will examine the parametrised enumeration complexity of the task to enumerate all strong $\mathcal{C}$-backdoor sets of a given 3-CNF formula for some clause-defined base class $\mathcal{C}$. Crucially, every strong backdoor set has to contain at least one variable from a clause that is not in $C$ which relates to "hitting all bad clauses" like in the definition of deletion backdoors (see Definition 15).

**Theorem 5.** *For every clause-defined base class $\mathcal{C}$ and input formula in 3-CNF:*

- ENUM-STRONG-$\mathcal{C}$-BACKDOORS$^{\text{LEX}}$ $\in$ DelayFPT *with polynomial space and*
- ENUM-STRONG-$\mathcal{C}$-BACKDOORS$^{\text{SIZE}}$ $\in$ DelayFPT *with exponential space.*

**Proof.** We show that for every clause-defined base class $\mathcal{C}$ and input formula in 3-CNF, the problem MIN-STRONG-$\mathcal{C}$-BACKDOORS is FPT enumerable. Indeed, we only need to branch on the variables from a clause $C \notin \mathcal{C}$ and remove the corresponding literals over the considered variable from $\phi$. The size of the branching tree is at most $3^k$. As for base classes the satisfiability test is in P, this yields an FPT algorithm. The neighbourhood function $\mathcal{N}(x, S)$ for $x = (\phi, k)$ is defined to be the set of the pairwise unions of all minimal strong $\mathcal{C}$ backdoors of $(\phi[(S \cup \{x_i\})], k - |S| - 1)$ together with $S \cup \{x_i\}$ for all variables $x_i \notin S$. If $\text{Vars}(\phi) = \{x_1, \dots, x_n\}$, then the operations are $\omega_i \colon \phi \mapsto \phi(0/x_i) \wedge \phi(1/x_i)$. As application of the functions $\omega_i$ happens only with respect to the backdoor size $k$, which is the parameter, the formula size increases by an exponential factor in the parameter only. This yields the preconditions for Corollary 7 . □

*4.3. Weighted Satisfiability Problems*

Finally, we consider satisfiability questions for formulas in the Schaefer framework [8]. A constraint language $\Gamma$ is a finite set of relations. A $\Gamma$ formula $\phi$, is a conjunction of constraints using only relations from $\Gamma$ and, consequently, is a quantifier-free first order formula.

As opposed to the approach of Creignou et al. [13], who examined maximum satisfiability, we now focus on the problem MINONES-SAT($\Gamma$) defined below.

If $\theta \colon X \to \{0, 1\}, \theta' \colon Y \to \{0, 1\}$ are two (partial) assignments over some set of variables $X$ and $Y$, then $\theta \subset \theta'$ is true, if $\theta(x) = \theta'(x)$ for all $x \in X$ and $X \subset Y$.

**Definition 18** (Minimality). *Given a propositional formula $\phi$ and an assignment $\theta$ over the variables in $\phi$ with $\theta \models \phi$, we say that $\theta$ is minimal if there does not exist an assignment $\theta' \subset \theta$ and $\theta' \models \phi$. The size $|\theta|$ of $\theta$ is the number of variables it sets to true.*

Formally, the problem of interest is defined with respect to any fixed constraint language $\Gamma$:

| **Problem:** | MIN-MINONES-SAT$^{\text{SIZE}}$($\Gamma$) |
| --- | --- |
| **Input:** | $(\phi, k)$, a $\Gamma$ formula $\phi$, $k \in \mathbb{N}$. |
| **Parameter:** | The integer $k$. |
| **Output:** | Generate all inclusion-minimal satisfying assignments $\theta$ of $\phi$ with $|\theta| \le k$ by non-decreasing size. |

Similarly, the problem ENUM-MINONES-SAT($\Gamma$) asks for all satisfying assignments $\theta$ of $\phi$ with $|\theta| \le k$. In this context, the operations in $\Omega$ are functions that replace the variable with index $i \in \mathbb{N}$ by true.

**Theorem 6.** *For all constraint languages $\Gamma$, we have:* MIN-MINONES-SAT$^{\text{SIZE}}$($\Gamma$) *is FPT enumerable and* ENUM-MINONES-SAT$^{\text{SIZE}}$($\Gamma$) $\in$ DelayFPT *with exponential space.*

**Proof.** For the first claim we can simply compute the minimal assignments by a straightforward branching algorithm: Initially, begin with the all 0-assignment, then consider all unsatisfied clauses in turn and flip one of the occurring variables to true. The second claim follows by a direct application of Corollary 7 . □

## 5. Conclusions

We presented FPT delay ordered enumeration algorithms for a large variety of problems, such as cluster editing, chordal completion, closest-string and weak and strong backdoors. An important point of our paper is that we propose a general strategy for efficient enumeration. This is rather rare in the literature, where algorithms are usually devised individually for specific problems. In particular, our scheme yields DelayFPT algorithms for all graph modification problems that are characterised by a finite set of forbidden patterns.

Initially, we focussed on graph-theoretic problems. Afterwards, the generic approach we presented covered problems which are not only of a graph-theoretic nature. Here, we defined general modification problems detached from graphs and constructed generic enumeration algorithms for arising problems in the world of strings, numbers, formulas, constraints, etc.

As an observation, we would like to mention that the DelayFPT algorithms presented in this paper require exponential space due to the inherent use of the priority queues to achieve ordered enumeration. An interesting question, continuing the research of Meier [45], is whether there is a method which requires less space but uses a comparable delay between the output of solutions and still obeys the underlying order on solutions.

**Author Contributions:** Conceptualization, N.C., R.K., A.M., J.-S.M., F.O. and H.V.; Funding acquisition, N.C., A.M. and H.V.; Methodology, N.C., R.K., A.M., J.-S.M., F.O. and H.V.; Supervision, N.C., A.M. and H.V.; Writing—original draft, N.C., R.K., A.M., F.O. and H.V.; Writing—review and editing, N.C., R.K., A.M., F.O. and H.V.

**Funding:** This research was funded by Deutsche Forschungsgemeinschaft (ME 4279/1-2) and the French Agence Nationale de la Recherche (AGGREG project reference ANR-14-CE25-0017).

**Acknowledgments:** We thank the anonymous reviewers for their valuable feedback.

**Conflicts of Interest:** The authors declare no conflict of interest.

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
