# Peer review of "Parameterised Enumeration for Modification Problemsâ€"

_algorithms, doi:10.3390/a12090189_

Round 1
Reviewer 1 Report
The paper introduces a general framework to discuss enumeration following a fixed order with FPT-type delay-bounds. The paper focusses on modification problems and gives some very general properties for which enumeration with FPT-delay is possible for lexicographic order and order by size (cardinality of the solution). Specifically this gives results for a large class of graph modification problems (including cluster editing, triangle detection,...) and also non-graph problems such as closest string and variations of satisfiability.
Although the authors emphasise their results on graph modification problems, I consider the very general study of modification problems to be the main contribution as it gives insight into the helpful structure that allows for DelayFPT enumeration for a very broad class of problems. General results of this form are rare, so I would suggest to mention this more prominently also in the abstract. Developing these results by looking at the special case of graph modification problems fist is a nice build-up and easy to follow. To me, the results on the graph modification problems were not too surprising, their generalisation however is the interesting consequence. The collection of specific problems to which this technique can be applied is indeed various but also leaves room for further research.
More specific remarks
I'd suggest to mention already in the introduction that your enumeration algorithms do not consider minimal solution, but solutions of a bounded size, and further that extending to such solutions from the minimal ones is not generally trivial. Coming from a different branch of enumeration algorithms, it took me reading half the paper to understand this difference.
The related work paragraph on page 2 is pretty small, perhaps it is better to move this into the third paragraph on this page, to where the framework for DelayFPT is discussed. This is afterall also related work. If the journal format specifically requires a distinctly highlighted section called 'related work' at the end of the introduction, a second mentioning of the framework and the results studied (reference 12) could give more context to the thin paragraph that feels a bit lost in its current state.
Conceptually I was a bit confused to see that the quasi order of a parameterised enumeration problem is defined on the union of all solutions. Sure, it is a quasi order which then only relates solutions that work for the same instance but at the point of the definition on page 3 this is not so obvious. To be honest, I expected a quasi-order which depends on the instance, especially since you later require (what is to be expected) that comparing two solutions only takes time dependent on the size of the input. Perhaps add a short explanation that this order on all solutions is only a lazy way of simultaneously giving an order for each instance.
The paragraph describing the priority queue (pages 3-4) reads very strangely. After mentioning reference 5, it is not clear what 'exponential' refers to. An initial statement like the following would give a proper context of what you want to describe: 'For an instance x of a parameterised enumeration problem, we use a priority queue Q to store a subset of Sol(x), of cardinality potentially exponential in |x|.' Further, I believe this part requires a bound on the size of a solution to an instance, not just the number of solutions in the run-times (at this point you are talking about general parameterised enumeration problems, not about the specific modification problems where this is bounded by |x|).
The paragraph starting p11l352 is pretty confusing, actually wrong in the stated context. In general, there is no equivalence of strong and deletion backdoor sets. The notion clause-defined occurs later, and this restriction also only yields classes for which deletion implies strong and not the reverse. I suggest to only define deletion backdoor sets in this paragraph on p11l352. Perhaps just before Theorem 5 one could mention that every strong set has to contain at least one variable from a clause that is not in C which relates to 'hitting all bad clauses' like in the definition of deletion backdoors. This gives rise to the branching used in the Theorem and gives a good intuition of what happens there.
Typos/Rephrasing
Decide between s and z for parameterised, parameterisation, etc.
Maybe it is just the style, but to me the quotation marks look wrong, for example ``cost'' in p1l21 and ``to be triangular'' p4l153.
p2l34 systematically are tried -> are tried systematically
p2l67 exists already -> already exists
p2l68 reference 8 should be reference 12
p3l82 .. that is, he has to produce solutions without violating that ordering. -> that is, it has to produce solutions without violating this ordering.
p5l181 for any graph modification problem M_P, if the decision problem -> for any graph property P, if M_P
p8 Algorithm 3: Why do you exclude only t from O and not S\cup {t} in line 3?
p9 in Definition 11: pair of permutation -> pair of permutations
p9 in Definition 11: what is S_k? I suppose you mean the symmetric group, but it looks very weird here, since S is always the solution set. Maybe "permutations on k elements" is enough as description.
p10l315 EnumMin -> Enum-Min
p10l341 which allow for efficient solving SAT -> for which satisfiability can be decided efficiently
p10l342 one tries to overcome a small distance of a formula of -> of a formula tries to describe a distance to tractability
p12l407 all assignments -> all satisfying assignments
Reviewer 2 Report
In the present article the authors study methods for enumerating
solutions to problems with a combinatorial flair. For example,
enumerating all models of a propositional formula, or enumerating all
graphs satisfying some given property. Two additional constraints are
considered: the given problem is
fixed-parameter tractable, and the enumeration of
solutions has to follow a given order (e.g.,
lexicographic). Particular focus is placed on graph modification
problems, where three general results are obtained (Lemma 1,
Theorem 1, Theorem 2). In Section 5 these results are extended and
presented in a more general framework, accompanied by three concrete
examples.
Compared to several existing results on enumeration algorithms, the
findings are rather general, and the proposed algorithms are
in principle applicable to a wide range of problems. However, the
algorithms are more or less straightforward combinations of well-known
ideas. This alone would not be a problem if the authors could
demonstrate that their approach could resolve some open cases in the
literature, or at least strongly generalised existing methods, but the
provided examples are all rather trivial, and there is no indication
that their enumeration status was ever in serious doubt. If my
understanding of this is incorrect, then the authors need to argue
this much more strongly in the introduction, and provide necessary
references.
Nevertheless, it should be mentioned that there are few general
complexity results concerning enumeration, and it is an area of
theoretical computer science which has traditionally received less
attention. In this light the findings of the article are perhaps more
valuable, and it would be good if this general formulation of
enumeration problems is published in a journal, for future
reference. However, the current structure of the paper needs to be
reworked. I would propose to remove Section 3 and Section 4 and
instead begin with the more general Section 5 (which needs to be
rewritten accordingly). Then the graph modification problems can either be
running examples or be placed in a subsection after the general
statements. In addition, the following list of comments should also be taken into
consideration.
* Throughout the article there are many occurrences of the word "some"
where it should be "a" or "an".
* Line 20: "high interest" sounds a bit biased.
* Line 88: the domain and image of the Sol needs to be clearly
specified.
* Line 102: "the time" needs to be made precise. Time with respect to
what?
* Line 129: you never define "graph property" properly.
* Line 130: the definition of graph operations is too informal.
* Lemma 1: the proof is too sketchy.
* Corollary 2: "Combine Proposition 1 with Lemma 1" should rather be
"This result follows by combining Proposition 1 with Lemma 1", or
something along those lines.
* Line 216: "In the following" -> "In the following,".
* Line 238: "output" -> "outputted".
* Line 268: "see" -> "show".
* Corollary 4: this proof is too sloppy.
* Definition 10: similar to the graph case you never actually define
what you mean by an "operation". In general terms an operation is
simply a k-ary from X to X, for some fixed set X, but this is not
what you want. Also, what do you mean by a "valid operation"?
* Example 1: removeVertex_i, removeEdge_{i,j}, and addEdge_{i,j} have
not been properly defined, as far as I can tell.
* Definition 11: you still haven't defined a "property".
* Line 310: I don't think "neighbourhood function" has been defined in
this context.
* Line 321: what is "d_H"?
* Theorem 3: rewrite into something more rigid.
* Line 341: rewrite this sentence.
* Line 355: "Denote with C be some class..." -> "Let C be a class...".
* Line 367: what do you mean by "can be recognised in P"?
* Line 378 and 408: "assignTruthValueToVariable(t,i)" and "setVariableToTrue(i)" do not make any
sense. It seems that you assume the existence of an undefined
programming language, and assume that the reader is familiar with all
these constructs.
* Line 384: end this line with ":".
* Line 397: I would prefer "relation" rather than "logical relation".
* Line 400: "where they" -> "who".
* Line 419: "where usually algorithms are..." -> "where algorithms are
usually..."
* References 12, 14, 19, 24, 33, 34, 38, 40, 41, and 44 should be expanded.
Reviewer 3 Report
Summary
The authors study a wide variety of enumeration problems and ask algorithms enumerating all solutions with respect to some order such that the delay between two solution outputs is bounded by either a polynomial or by some FPT-function. The main results are a) a general tool/algorithm that works whenever the underlying enumeration problem (with no requirement on the order of the solutions) can be solved in FPT-time and b) an algorithm that works whenever one can find a "neighbourhood function", that is, a function that (roughly speaking) defines the neighborhood of a solution as new solutions that are larger in size and such that each solution is in a finite neighborhood-radius from some initial solution.
Good things about the paper
The paper is well written, the proofs are (except for one) easy to follow and the new tool seems to be quite nice for proving membership in FPT-Delay even if the solutions have to sorted.
Things that should be improved in the paper
1) The relevance of and motivation for the paper are not clear to me and should be addressed more carefully.
2) The authors switch between British English and American English (see eg. parametrisation, parameterisation, parameterization, characterisation, optimization, capitalization)
3) All results are pure classification results. I would prefer to know the actual running times of the algorithms.
4) It is not really clear whether the results are new (or surprising) or whether the results are known but the proofs are new and simpler.
Final thoughts
The paper and its results are sound but not really technically involved. The motivation for this paper is too thin for me to be fully conviced that the paper should be acceptance. For this reason I would advise a revision of the paper.
Specific Remarks
Is it "pôle technologique de Sfax", "Pôle technologique de Sfax" or "Pôle Technologique de Sfax"?
There is a missing "r" in the email address of the third author.
l19: "is of utmost importance" is a bit strong for my taste given that the motivation is not super convincing.
l25: Is it easy to see why this method cannot guarantee this?
l26: "to any constraint" seems to be too strong. What about properties of constraints that are not even decidable?
l32: Do you mean "for any other classes"?
l63: What do you mean by "exponential space"? This the algorithm runs in FPT-time, it can only use FPT-space, right? Then why not use FPT-space instead of exponential?
l82: The "he" refers to the algorithm, correct? Then it should be "it".
l86: Is there a reason why you work with sets of instances instead of just instances?
l87: This implies that the tree-width of a graph is not a parameterization?
l90: Is it a wanted feature that the solutions are not ordered by their respective instance?
l98: Formally, z "not at most" y is never defined.
l124: Section 3 seems to only continue the Preliminaries. Why have a new Section then?
l136: When are two graphs equal? One could understand isomorphic here and then the claim would be incorrect as eg. deleting any edge from a triangle results in a P_3.
l154: "can be problematic" - aren't they always?
l155: "as well as we will see later" - I think you mean "as we will see later".
l169: Is there a reason to count the bits from right? Just change "last" in l3 of Alg1 to "first".
Algorithm 1: I think the overload of O is confusing and unnecessary. "Let O' := O without o_p"...
l226: What is "k"? I guess you mean "kappa(x)".
l235 & 245: "runs in DelayFPT" - DelayFPT is a complexity class. I think it should be "runs in DelayFPT-time".
l257.5: Please put Algorithm 3 to the top of the page.
l261 & 267: What do you mean by "expected sequence"?
l272 & 276: Can they produce all inclusion-minimal solutions or all solutions of minimum size?
l288: "That being so" sounds strange.
l289: If "i,j\in V", then i and j do not encode vertices but are vertices.
Definition 11: What is "i_mu"
l297: I would delete the "(of course)".
l304 and at other parts: You are inconsistent with "Parameter: k." and "Parameter: The integer k \in N.".
l309 ff: Where are the proofs of Corollaries 5 and 6? Since they are referenced a lot later I would really like to check their correctness.
l327: The input is hard to parse. A suggestion: "A set {s_1,s_2,...,s_k} of k strings over {0,1} each of length n\in N and an integer d\in N."
l347: "for a truth assignment tau" - I would add "partial" to indicate that tau does not need to set all variables.
l352-254: What about clauses of the form "x or not x or t" where t is not in C? It is only a special case but I think the observation does not hold for these.
l355: There is an additional "be" in "Denote with C (be) some class"
l360: "one defines" sounds strange.
l375: I think that "3-CNF" is much more common than "3CNF".
l378-380: I do not get this proof. Please clarify.
l394-395: I do not get this sentence.
l401: I thought that orderings are the novelty of this paper and [12] focuses on elimination of duplicates.
l403: What do you mean by "strictly less"? I would interpret it as "a smaller number of" and not as "a strict subset". In the problem definition it is "inclusion-minimal".
l406: Why is the problem called Min-MinOnes-SAT^SIZE? Are there two things that are minimized? If I understand it correctly, then only the number of variables set to true is minimized. (And the solutions are ordered by size.)
l422: I do not think that this paper really focuses on graph problems.
l431ff: What is the difference between Conceptualization and Methology? Similar: Since the list of contributers is the same for "Writing - original draft" and "Writing - review and editing" wouldn't it make sense to merge them?
l489: Is there a reason that this is the only reference with an URL?
Round 2
Reviewer 1 Report
My previous comments have been addressed to my satisfaction, aside from minor phrasing issues/typos listed below. The overall switch of focus towards the results for a general framework give the paper a much better presentation. Some edits introduced one, to me, unpleasant double definition of "consistency".
More precisely, for the general modification problems, the notion of consistency is defined twice, once for sets of operations and then again and differently for solution sets in particular, in Definitions 10 and 11. Since you are only interested in consistent solutions anyway, why not use the term consistency for operations sets which satisfy both the properties "order does not matter" and "operations to not cancel each other out" (written more formally, of course), in Definition 10, where operation sets are introduced. Then in Definition 11, you can remove the messy "there exists an ordering" part, and simply define a \emph{solution} to be a consistent set of operations which translates the instance to a word in P. In the problem definition, you can also remove $S(x)\in P$, as this is implied by the property of $S$ being a solution.
Typos
p2l54 "further push the topic" sounds a bit too informal for me... How about "support this topic"
p2l63 "do not start from..." is not what you mean, replace the according sentence by "Notice that the presented algorithms do not enumerate the set of minimal solutions but the set of solutions of bounded size." (Otherwise, I am very pleased with these additional sentences here!)
p3l99 I see you literally included my sloppy "lazy way" comment, this is maybe a bit too honest here :) How about "short way" instead?
p4l135 modifications -> modification
p4l141 Move the first sentence of Definition 6 to the sentence above that defines graph property.
p4l149 and l151: replace "one" and "the other" by $o$ and $o'$, respectively.
p7l252 Don't you need FPT-computability for all S in Sol AND the set {O} ?
p9l296 size minimal -> inclusion-minimal (or am I missing something here?)
p9 Proof of Corollary 5:
Regarding -> regarding
delete "obviously just"
Finally, just apply -> The result then follows from
p10l324 emphasize the word "property"
p10l326 was -> is
p11l374 whose maximum Hamming distance -> with
p11l376 and p12l406 obtained ... in -> obtained ... by
p13 Proof of Theorem 4
delete all three occurrences of "then"
l431 sets -> set
p13 Definition 18: I liked the previous definition of minimality, the new one with the undefined inclusion symbol for assignments seems confusing to me. Perhaps another reviewer disagrees... In this case, you should define the inclusion symbol for assignments somewhere.
Reviewer 2 Report
The revised version contains many improvements over the original article. My recommendation is to accept the article if the following remarks are sufficiently addressed:
* Line 20: "often it is central" -> "it is often central".
* Line 32: "for any other classes" should probably be "for any other
class"
* Line 68: "general modification problem" is not a widely understood
term. You should at least give the reader some intuition of what such
problems can be.
* Line 98: I don't understand what you mean by "a lazy way of simultaneously giving an order for
each instance".
* Line 126: I would remove "the concept of" from this sentence.
* Line 165: "The integer k" should be "An integer k".
* Line 209: "runts" -> "runs".
* Line 306: "Regarding" -> "regarding".
* "Finally, just apply Corollary 4" -> "Finally, we apply Corollary
4".
* Line 312: "and x..." -> "and let x...".
* Line 314: "We say an operation..." -> "We say that an operation...".
* Line 314: "We write ... as the set of..." -> "We write ... for the
set of...".
* Line 318: end this item with a period.
* Line 320-321: I don't understand the sentence "It highly is the
subject to the repective language Q.". Also, "repective" should be
"respective".
* Line 325: "you may think" is out of place. I would prefer to replace
this sentence with "Intuitively, in the context of graph modification
problems, we may think of Q as G.", or something similar.
* Line 326: "was just a subset" -> "is just a subset".
* Line 340: "regarding property P over Sigma" -> "with respect to a
property P over Sigma". The same remark holds for the definition of
Enum-Min-Pi_P as well.
* Line 379: ".." -> ".".
* Line 387: This proof can be improved. For example, "Then use
Proposition 2 and Corollary 7" should be written as "By combining
Proposition 2 with Corollary 7 we get the desired result." or
something along those lines.
* Line 451: I don't understand what you mean by "constituting the
proof".
* Abbreviations in the bibliography still needs to be standardised.
Reviewer 3 Report
Since the authors addressed all of my concerns from the first revision, I have only very minor comments for the second revision. I do recommend acceptance of the paper independent of whether the authors want to implement changes on the few remaining comments or not. These comments should therefore be seen as clarifications to former comments or minor beautification suggestions.
Concerning the two comments that were not clear to the authors:
E.g. corollary 7 shows a statement of the form: "If ENUM-MIN-Π_P is FPT-enumerable and consistency of
solutions can be checked in FPT then E NUM-Π_P ∈ DelayFPT and polynomial space or exponential space."
I would prefer a statement of the form "If ENUM-MIN-Π_P can be solved in f(kappa) * n^c time and consistency of
solutions can be checked in g(kappa) * n^d time then ENUM-Π_P can be solved with h(kappa) * n^e delay and O(A) space."
Of course with specific values for h, e, and A depending on f,c,g and d.
In Algoritm 1, in line 3 you have "O := O \ {o_p}". Does O now contain o_p or not? I know that this is really common in programming but in papers it makes it a bit clearer to have "O' := O \ {o_p}". Now it is clear that O does contain o_p while O' does not.
l93: For enumeration problems, we are usually given e.g. a graph G and the task is to find all motifs of a special form in G. Your approach is different: You are given a set of graphs and want to find all special motifs in all of the input graphs. While there is nothing wrong with it per se, it raises the question why you do it in this way. (Both ways are in some sense equivalent as one could give as input a set that only contains G or in the other direction one could ask all graphs from a set one after another.)
l436: There is still a 3CNF (consistent would be 3-CNF).
